# Potential Association of *Cutibacterium acnes* with Sarcoidosis as an Endogenous Hypersensitivity Infection

**DOI:** 10.3390/microorganisms11020289

**Published:** 2023-01-22

**Authors:** Yoshinobu Eishi

**Affiliations:** Department of Human Pathology, Graduate School, Faculty of Medicine, Tokyo Medical and Dental University, Tokyo 113-8510, Japan; eishi.path@tmd.ac.jp

**Keywords:** *Cutibacterium acnes*, *Propionibacterium acnes*, sarcoidosis, granuloma, pathogenesis, latent infection, reactivation, cell-wall-deficient, endogenous infection, immune complexes

## Abstract

The immunohistochemical detection of *Cutibacterium acnes* in sarcoid granulomas suggests its potential role in granuloma formation. *C. acnes* is the sole microorganism ever isolated from sarcoid lesions. Histopathologic analysis of some sarcoid lymph nodes reveals latent infection and intracellular proliferation of cell-wall-deficient *C. acnes* followed by insoluble immune-complex formation. Activation of T helper type 1 (Th1) immune responses by *C. acnes* is generally higher in sarcoidosis patients than in healthy individuals. Pulmonary granulomatosis caused by an experimental adjuvant-induced allergic immune response to *C. acnes* is preventable by antimicrobials, suggesting that the allergic reaction targets *C. acnes* commensal in the lungs. *C. acnes* is the most common bacterium detected intracellularly in human peripheral lungs and mediastinal lymph nodes. Some sarcoidosis patients have increased amounts of *C. acnes*-derived circulating immune complexes, which suggests the proliferation of *C. acnes* in affected organs. In predisposed individuals with hypersensitive Th1 immune responses to *C. acnes*, granulomas may form to confine the intracellular proliferation of latent *C. acnes* triggered by certain host-related or drug-induced conditions. Current clinical trials in patients with cardiac sarcoidosis are evaluating combined treatment with steroids and antimicrobials during active disease with continued antimicrobial therapy while tapering off steroids after the disease subsides.

## 1. Introduction

Sarcoidosis is a systemic disease with unknown etiology in which noncaseating epithelioid cell granulomas are formed in various organs. The disease is thought to be caused by the exposure of susceptible hosts to a specific causative agent triggered by environmental factors [1]. This proposed etiology of sarcoidosis involves three main factors: some pathogens as causative agents, host factors leading to disease susceptibility, and environmental factors that trigger the onset of the disease. Even if microorganisms are involved, sarcoidosis is not a simple infectious disease. While many infectious agents have been investigated, the causative agent is not yet established. For example, researchers in Western countries long suspected *Mycobacterium tuberculosis* as a causative agent, but no acid-fast bacilli (including *M. tuberculosis*) have been cultured from sarcoid lesions [2]. Further, QuantiFERON-TB Gold test results do not differ significantly between peripheral blood mononuclear cells (PBMCs) obtained from sarcoidosis patients and healthy controls [3,4,5,6].

## 2. Isolation of Microorganisms

In Japan, a research project supported by the Japanese government was organized in 1978 to investigate the cause of sarcoidosis. Tissue suspensions from biopsied lymph nodes were cultured in various media, inoculated into sterile mice and nude mice, and then each organ of the mice was cultured in the same way. The investigation began by excluding all bacterial, viral, and fungal organisms one by one. *Cutibacterium acnes* (formerly *Propionibacterium acnes*) was isolated from 78% of the 40 cases of sarcoidosis, and no other microorganisms (including *M. tuberculosis*) were isolated [7,8]. The isolation frequency of *C. acnes* increased to 92% when using improved culture media (high osmolarity culture media), and almost all *C. acnes* cultures from patients with active sarcoidosis were successful. Compared with sarcoidosis patients, the isolation frequency of *C. acnes* in biopsied lymph nodes from control patients without sarcoidosis was significantly lower (25% of 150 cases), and fewer isolated colonies were obtained. Based on these results, the potential association of *C. acnes* with sarcoidosis was proposed in Japan, but many unresolved issues remain.

## 3. Pathology of Granuloma Formation

Granuloma formation is a biologic defense response to eliminate and isolate foreign substances indigestible by cells. The two mechanisms of granuloma formation—persistence of a nondegradable product and hypersensitivity responses—overlap in most infectious diseases because microorganisms act as both foreign bodies and antigens to induce immunologic responses [9]. T helper type 1 (Th1) immune responses induce epithelioid transformation of granuloma cells. Epithelioid granuloma cells have greater digestive ability than conventional macrophages and can degrade or abolish the causative agents [10,11]. A classical pathologic principle in diagnosing granulomas is that the causative agent must locate within the granuloma.

## 4. The Kveim Reaction

In the Kveim test, the presence of sarcoid granulomas is evaluated several weeks after the intracutaneous injection of syngeneic or allogeneic sarcoidosis lymph node or spleen tissue homogenates [12,13]. Because injection of the Kveim reagent induces sarcoid granulomas, the triggering agent of sarcoid granulomas was thought to be an ingredient of the Kveim reagent [14]. The Kveim reaction is consistent with an antigen-specific cellular immune response characterized by an increase in CD4+ T cells and histiocytes with oligoclonal T-cell expansion at the site [15]. In patients without sarcoidosis, the results of the Kveim test are negative [16], which suggests that the antigen-specific immune responses to the triggering agent of sarcoid granulomas may be specific to sarcoidosis patients [14].

## 5. *C. acnes* in Sarcoid Granulomas

A search for the causative agent of sarcoidosis via immunohistochemistry was performed based on two premises: the presence of the causative agent in sarcoid lymph nodes based on the Kveim reaction and localization of the causative agent in sarcoid granulomas based on the pathologic principle of granuloma formation [17,18]. In early studies, the SG5 antibody, which reacts with an exogenous antigen located in sarcoid granulomas, was generated by immunizing mice with sarcoid lymph node tissue homogenate followed by immunohistochemical screening of antibody-producing hybridoma clones using formalin-fixed paraffin-embedded (FFPE) sarcoid lymph nodes [17]. The SG5 antibody reacted specifically with a *C. acnes* culture supernatant and not with other bacterial supernatants (including those of *M. tuberculosis*). Accordingly, an anti-*Propionibacterium acnes* monoclonal antibody (PAB antibody) that reacts with a species-specific lipoteichoic acid (LTA) of *C. acnes* in sarcoid granulomas was developed by immunizing mice with the whole bacterial lysate and conducting immunohistochemical screening of *C. acnes*-specific antibody-producing hybridoma clones using FFPE sarcoid lymph nodes [18].

Immunohistochemistry with the *C. acnes* LTA-specific PAB antibody revealed positive signals in sarcoid granulomas in 88% of sarcoid lymph nodes and 74% of sarcoid lungs; no positive signals were detected in non-sarcoid granulomas in cases with tuberculosis or sarcoid reaction [18]. Positive PAB antibody signals were also observed in sarcoid granulomas obtained from originally aseptic organs such as the heart [19] and eyeball [20,21]. Localization of the immunohistochemical signals to *C. acnes* in sarcoid granulomas is shown in Figure 1. Numerous case reports have described *C. acnes* detection in the granulomas of patients with pulmonary sarcoidosis [22,23,24,25], cutaneous sarcoidosis [26,27,28,29,30,31], nasal sarcoidosis [32], and neurosarcoidosis [33,34]. The term “*C. acnes*-associated sarcoidosis” is applied to cases in which *C. acnes* is detected in granulomas via immunohistochemistry using the PAB antibody [35]. The detection sensitivity of *C. acnes* in sarcoid granulomas depends on the evaluation method [36,37]; an automated method of detection using a Leica system with a commercially available PAB antibody (MBL, D372-3) has been used for differential diagnosis of sarcoidosis from other granulomatous diseases.

## 6. Histopathologic Analysis of *C. acnes* in Sarcoid Lymph Nodes

PAB-antibody-positive structures in sarcoid granuloma cells are observed via electron microscopy as small round electron-dense bodies that cannot be identified as bacteria by their morphology alone because the cell wall structure is lacking (Figure 2) [38]. In the early stage of granuloma formation, macrophages are filled with PAB-antibody-positive small round bodies, and some are accompanied by PAB-antibody-positive Hamazaki-Wesenberg (HW) bodies with a large spindle shape. A report published in 1966 described HW bodies in association with sarcoidosis [39]. These HW bodies had ceroid characteristics [40] and were suspected to be cell-wall-deficient forms of *M. tuberculosis* [41,42].

Electron microscopy shows small round bodies extruding from the HW bodies in macrophages that are indicative of a dividing cell-wall-deficient bacterium [18]. Immunoelectron microscopic analysis of HW bodies using two *C. acnes*-specific antibodies (PAB-antibody-detecting cell-membrane-bound LTA and TIG-antibody-detecting ribosome-bound trigger factor protein) revealed that the PAB antibody signals distribute around the periphery of the HW bodies, whereas TIG antibody signals distribute in a dot-like pattern over the entire internal region of the body [18]. The distribution pattern of these bacterial components in the HW body structure is consistent with their localization in the basic structure of a bacterium despite the lack of a cell wall structure, which indicates that the HW bodies themselves are the bacterial bodies of cell-wall-deficient *C. acnes* (Figure 3). HW bodies detected by these *C. acnes*-specific antibodies are mainly located in sinus macrophages of the lymph nodes and, although not specific to sarcoidosis patients, are present at a significantly higher frequency in sarcoid than in non-sarcoid samples (50% of 119 cases vs. 15% of 165 cases, respectively) [18].

## 7. Latent Infection and Intracellular Proliferation of *C. acnes*

Based on histopathologic analysis, HW bodies are thought to represent latently infected *C. acnes*, while the small round bodies filling the cells are thought to represent intracellularly proliferating *C. acnes*. Unlike extracellular cell-walled *C. acnes* in the hair follicles of the skin, the intracellular *C. acnes* lack a cell wall. In culture medium containing a rich nutrient, these intracellular cell-wall-deficient *C. acnes* seem to revert to a conventional cell-walled form. Therefore, isolation of *C. acnes* from sarcoid lesions requires a longer incubation period than usual and a highly osmotic culture medium. In the early stages of culture, the organisms often appear round rather than coryneform.

A meta-analysis of 58 studies that included more than 6000 patients from several countries to investigate all types of infectious agents proposed to be associated with sarcoidosis revealed that *C. acnes* is most commonly linked to sarcoidosis [43]. Intracellular proliferation of *C. acnes* in sarcoid lesions was demonstrated via quantitative polymerase chain reaction (PCR) using FFPE lymph node specimens [44,45]. Although *C. acnes* DNA can also be detected in some control samples, sarcoid samples contain a larger number of *C. acnes* genomes—almost the same number of *M. tuberculosis* genomes detected in tuberculosis samples [44]. In situ hybridization using signal amplification with catalyzed reporter deposition also shows the localization of large numbers of *C. acnes* in sarcoid granulomas [46]. Lymph node samples from European patients with sarcoidosis examined in an international collaboration study [47] also contained high amounts of *C. acnes* DNA, but *M. tuberculosis* DNA was almost undetected, which suggested the potential association of *C. acnes* with sarcoidosis even in European countries where *M. tuberculosis* has long been the suspected cause of sarcoidosis.

## 8. Humoral and Cellular Immune Responses to *C. acnes*

*C. acnes* is a commensal Gram-positive anaerobic bacterium that lives extracellularly on the skin and mucosal surfaces of the oral cavity and the gastrointestinal and genitourinary tracts [48]. The number of *C. acnes* detected via quantitative PCR in sebaceous materials aspirated from normal hair follicles begins to increase after the age of 10 years and reaches a maximum at the age of 15–19 years [49]. Serum antibody titers against *C. acnes* LTA (humoral immune response) are raised in all adults with no significant difference between healthy controls and sarcoidosis patients [50]. In contrast to the humoral immune response, PBMCs obtained from sarcoidosis patients that are stimulated by viable *C. acnes* exhibit increased production of a Th1-type cytokine interleukin-2 (cellular immune response) compared with PBMCs obtained from healthy controls [51]. Cellular immune responses are similarly increased when recombinant proteins of a *C. acnes* trigger factor [52] or catalase [53] are used as stimulating antigens. Bronchoalveolar lavage cells from sarcoidosis patients exhibit T-cell responses to even heat-killed *C. acnes* [54,55,56]. These observations suggest that sarcoidosis patients are predisposed to allergic cellular immune responses to this commensal bacterium.

## 9. Allergic Cellular Immune Response to Autoantigens

Granuloma formation may occur only in so-called “high responders” to *C. acnes*. The assumption of such a host factor is also based on the phenomenon of the Kveim reaction. Allergic cellular immune responses in sarcoidosis patients are assumed on the basis of experimental models of organ-specific autoimmune diseases such as experimental autoimmune encephalomyelitis [57] and experimental autoimmune thyroiditis [58]. For example, in an animal model of experimental autoimmune thyroiditis, the subcutaneous immunization of healthy animals with thyroglobulin using complete Freund’s adjuvant induces lymphocytic thyroiditis that simulates Hashimoto’s disease. In this animal model, DA strain rats are high responders and PVG strain rats are low responders; serum antibody titers to thyroglobulin are elevated in both DA and PVG rats, but lymphocytic thyroiditis occurs only in DA rats [59]. Such disease susceptibility of the host’s immune system [60] and dissociation of humoral and cellular immune responses to an identical antigen are also observed in sarcoidosis patients if *C. acnes* is assumed to be an intracellular antigen like the organ-specific autoantigen. Furthermore, experimental autoimmune thyroiditis is always self-limiting when induced in healthy animals, and DA rats that have undergone spontaneous remission are resistant to re-induction of experimental autoimmune thyroiditis [61]. These experimental observations are consistent with clinical observations that many sarcoidosis patients experience spontaneous remission.

## 10. An Experimental Model of Sarcoidosis Caused by *C. acnes*

Similar to the experimental models of organ-specific autoimmune diseases, allergic cellular immune responses to *C. acnes* can be induced in healthy mice via subcutaneous immunization with *C. acnes* using complete Freund’s adjuvant. Some mice develop multiple granulomas in the lungs, which suggests that the target antigens are present in the peripheral lungs where the granulomas are formed [62,63]. The frequency of mouse pulmonary granulomatosis ranges from 25%–57%, and the latent infection rate of *C. acnes* in mouse lungs is 33% by culture [63]. The administration of antimicrobials before and during the experiment can suppress the experimentally induced pulmonary granulomatosis [38,62]. These observations suggest two groups of mice: those with and those without latent infection of *C. acnes* in the peripheral lungs. In mice with latent infection, immunization with *C. acnes* using complete Freund’s adjuvant induces allergic cellular immune responses to *C. acnes*, and multiple foci of granulomatous inflammation are induced by targeting *C. acnes* that is infecting the lungs. On the other hand, in mice without latent infection, granulomatous inflammation does not occur even if an allergic reaction is induced because there is no target antigen in the lungs. In humans, *C. acnes* is cultured from peripheral lung tissue and mediastinal lymph nodes in adults at a frequency of 50–60% [64].

## 11. *C. acnes*-Derived Immune Complexes

*C. acnes* forms immune complexes extracellularly due to the presence of infectious antibodies raised by commensal *C. acnes* in the body. Many IgA/IgM-bound insoluble immune complexes derived from *C. acnes* in sarcoid lymph nodes are found in macrophages located in hyperplastic lymphatic sinuses [65]. Both the PAB antibody used for immunohistochemical detection of *C. acnes* and the infectious antibody against *C. acnes* LTA bind to an identical epitope of LTA on the bacterial surface. Due to epitope competition, PAB antibodies are unable to react with the immune complexes bound with immunoglobulins. After removal of the bound immunoglobulins via trypsin digestion, PAB antibodies can detect *C. acnes*-derived immune complexes [65]. Unlike intracellular latent or proliferating *C. acnes* that are mostly free from immunoglobulins, these immune-complexed *C. acnes* are originally extracellular *C. acnes* phagocytosed by macrophages (Figure 4): when the host cell is destroyed by intracellular *C. acnes* proliferation, *C. acnes* spreads outside the cell and is then captured by antibodies in the body fluids, phagocytosed as immune complexes, and digested intracellularly. If some survive intracellularly, a new latent infection may occur [66]. Unlike multivalent IgA/IgM, monovalent IgG does not easily aggregate the formed immune complexes. The IgG-bound immune complexes that escape local phagocytosis by macrophages may enter the lymphatic system and bloodstream. Indeed, in some biopsy samples of sarcoidosis, PAB-antibody-positive *C. acnes* is found in endothelial cells of lymphatic and blood vessels [38].

*C. acnes*-derived circulating immune complexes in the blood can be measured using a unique method to avoid epitope competition between the detection antibody and serum antibodies, although the method cannot directly identify the *C. acnes* antigen bound with immunoglobulins [50]. In many sarcoidosis patients, *C. acnes*-derived circulating immune complexes in the blood are abnormally increased. Such an antigenemia caused by *C. acnes* is likely to be caused by the reactivation and proliferation of latent *C. acnes* in the affected organs and may contribute to nonspecific systemic symptoms in sarcoidosis patients, such as fatigue [67].

## 12. Pathogenesis of Sarcoidosis Caused by *C. acnes*

The assumed pathogenesis of sarcoidosis caused by *C. acnes* is summarized in Figure 5. Commensal extracellular *C. acnes* causes asymptomatic intracellular infection via the respiratory tract. *C. acnes* is the most common bacterium detected intracellularly in the peripheral lungs and mediastinal lymph nodes in humans [18,36,64]. Susceptibility to latent *C. acnes* infection may be partly influenced by the NOD1 allele type of the host [68] or by catalase expression of the bacterium [69]. Latent *C. acnes* can be reactivated and proliferate intracellularly after certain triggering events, and this is not specific to sarcoidosis patients [18,36,70]. Autophagy is induced by an intracellular overload of *C. acnes* [71]; thus, intracellular *C. acnes* proliferation may induce the housekeeping function of autophagy, which plays a decisive role in determining the outcome of infection and immunologic balance [72]. It was recently proposed that dysfunction of mTOR, Rac1, and autophagy-related pathways not only hampers pathogen or nonorganic particle clearance, but also participates in T-cell and macrophage dysfunction, thereby driving granuloma formation [73]. The proposed mechanisms may contribute to the outcome of *C. acnes* infection and immunologic balance against intracellular *C. acnes* proliferation. Granuloma formation occurs only in individuals predisposed to a hypersensitive Th1 immune response against the intracellular proliferation of *C. acnes*. An allergic reaction to intracellular *C. acnes* proliferation seems to be caused by a different mechanism than the immunomodulatory effect of *C. acnes* itself [74]. Successful confinement of *C. acnes* that is proliferating intracellularly via granuloma formation prevents further spread of infective *C. acnes* to other cells, which resolves the granulomatous inflammation that leads to spontaneous remission in many sarcoidosis patients. Extracellular *C. acnes* that escapes granulomatous confinement or local phagocytosis has the potential to cause new latent infection in extrapulmonary organs (primarily in vascular endothelial cells) via dissemination through the lymphatic system and bloodstream, whereas local entry and subsequent latent infection of *C. acnes* through other than systemic spread may occur in some organs or patients. Latent infection in systemic organs can be simultaneously reactivated by additional triggering events, which leads to granuloma formation at all sites of latent infection in each organ of sarcoidosis patients.

## 13. Endogenous Hypersensitivity Infection Caused by *C. acnes*

Diseases caused by commensal microorganisms are referred to as endogenous infections and are generally classified into three major types (opportunistic infection, hypersensitivity, or the both-mixed type [38]) according to a classification system for microbial pathogens based on their ability to cause tissue damage as a function of the host’s immune response that was proposed by Casadevall and Pirofski [75]. In all three types, no lesion is induced under normal immune conditions. Sarcoidosis caused by *C. acnes* is classified as an endogenous hypersensitivity infection; granulomatous inflammation occurs only in patients with hypersensitivity responses against *C. acnes*. Diseases caused by a fungus such as candida or aspergillus are commonly encountered as opportunistic infections, but these microorganisms can also cause hypersensitivity pneumonitis with a background of a hypersensitivity predisposition and are thus classified as a both-mixed type endogenous infection.

Three major conditions must be fulfilled for sarcoidosis to develop as an endogenous hypersensitivity infection caused by *C. acnes*. The first prerequisite is latent infection by cell-wall-deficient *C. acnes*, which occurs not only in macrophages and vascular endothelial cells but also in epithelial and mesenchymal cells [76,77,78]. The onset of sarcoidosis is triggered by reactivation of latent *C. acnes*, which may be caused by host-related conditions such as physical or mental stress [79] or drug-induced conditions [80] such as the administration of interferons [25] and tumor necrosis factor-alpha antagonists [23]. The most critical cause of sarcoidosis is the host factor represented by Th1 hypersensitivity to the intracellular proliferation of latent *C. acnes*. A hypersensitive Th1 immune response to *C. acnes* may be caused by a disruption of peripheral T-cell tolerance to certain *C. acnes* antigens (as in organ-specific autoimmune diseases [81] or drug-induced sarcoidosis during administration of an immune checkpoint inhibitor [80]). The host–commensal relationship is supported by a unique regulatory T-cell population that mediates tolerance to bacterial antigens during a defined developmental window [82,83]. Thus, the predisposition of sarcoidosis patients with hypersensitivity to *C. acnes* may be influenced by how the developing immune system is exposed to this commensal bacterium.

Sarcoidosis as an endogenous hypersensitivity infection develops only after these three factors are established; that is, latent infection by *C. acnes* as the pathogen, reactivation of latent *C. acnes* triggered by environmental factors, and hypersensitive Th1 immune responses against the intracellular *C. acnes* as host factors.

## 14. Treatment Strategies for Refractory Sarcoidosis

Approximately two-thirds of sarcoidosis patients experience spontaneous remission within 2 years [84]. Sarcoidosis patients with organ damage or repeated relapses require treatment. According to the assumed pathogenesis, the intracellular proliferation of *C. acnes* causes granuloma formation and may provoke a new latent infection at the site of disease activity. Unless this latent infection is eliminated, recurrent inflammation can occur. With each recurrence, the affected tissue is destroyed by granulomatous inflammation in addition to postinflammatory fibrosis, resulting in gradual expansion of the lesion. Steroids have a therapeutic effect by suppressing allergic reactions. New granuloma formation can be prevented if the intracellular proliferation of *C. acnes* can be prevented by antimicrobial therapy. Tetracyclines, which are used for acne vulgaris [85], are effective against cutaneous sarcoidosis [86,87]. Although nonantibiotic properties of tetracyclines are suspected [88], a relationship between the antibiotic efficacy of minocycline for cutaneous sarcoidosis and the presence of *C. acnes* is suggested [89]. Antimicrobial agents may have limited efficacy against latent *C. acnes* infection (as in latent tuberculosis infection [90]). Because granuloma formation potentially prevents the extracellular spread of *C. acnes*, steroid administration to suppress granulomatous inflammation risks causing new latent infections. Therefore, the optimal treatment would be a combination of steroids and antibacterial therapy during periods of active disease with continued antibacterial administration while reducing the dose of steroids after the disease has subsided. Combination therapy with steroids and antimicrobials is currently under clinical trial investigation in Japanese patients with cardiac sarcoidosis [91]; the efficacy of the combination therapy is being compared between a control group receiving only steroids according to the current guideline [92] and an antimicrobial group receiving doxycycline and clarithromycin in addition to the steroids. A clinical trial of anti-*C. acnes* antimicrobial therapy using doxycycline and azithromycin in Dutch patients with sarcoidosis is also currently underway to compare the efficacy of the antimicrobial therapy between patients with or without detection of *C. acnes* with the PAB antibody in their biopsy samples [74]. In the study, the automated immunohistochemical method used to detect *C. acnes* with the PAB antibody should be standardized, however, because the sensitivity differs remarkably between the Leica and Ventana systems [36].

## 15. Conclusions

The results of many studies suggest the potential association of *C. acnes* as an endogenous hypersensitivity infectious agent that causes sarcoidosis. The concept of endogenous hypersensitivity infections is not limited to the etiology of sarcoidosis but should be considered as a general theory when investigating the etiology of other intractable diseases suspected to be caused by certain infectious microorganisms.

## Figures and Tables

**Figure 1 microorganisms-11-00289-f001:**
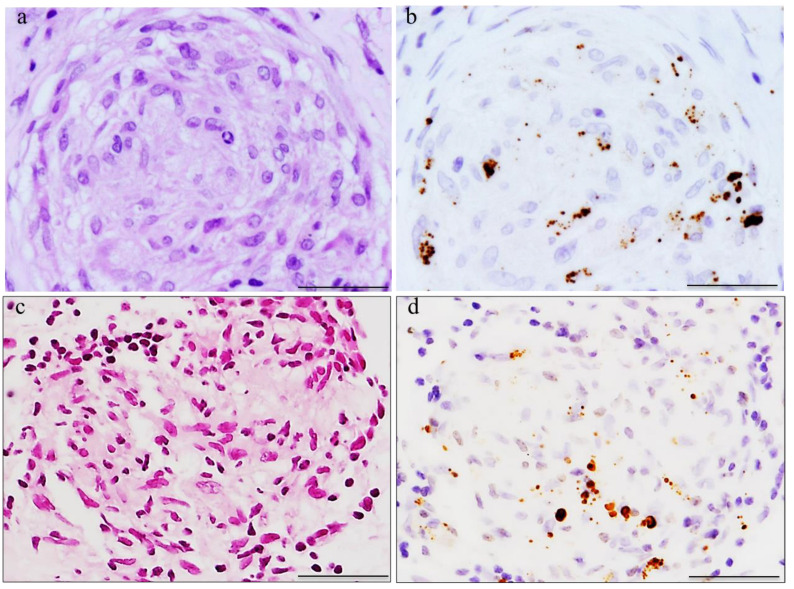
Immunohistochemical localization of *C. acnes* in sarcoid granulomas. Hematoxylin–eosin stain and immunohistochemistry with the *C. acnes*-specific PAB antibody are shown pairwise. Mainly small round and occasionally large ovoid PAB antibody-positive signals can be observed in non-caseating epithelioid cell granulomas of the lung (**a**,**b**) and ocular epiretinal membrane (**c**,**d**) from patients with sarcoidosis irrespective of the sites at which the granuloma formed. All photos are original and were previously published [35]. Scale bar: 50 μm.

**Figure 2 microorganisms-11-00289-f002:**
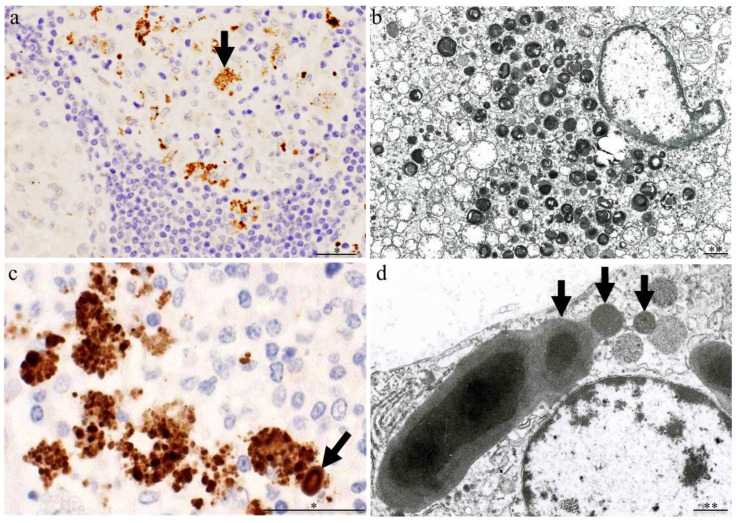
Features of intracellular *C. acnes* proliferation in sarcoid lymph nodes. Some granuloma cells (arrow) (**a**) are filled with PAB antibody-positive signals that are observed via electron microscopy as small round electron-dense bodies with a lamellar structure due to their partial deterioration (**b**). A cluster of swollen macrophages filled with many PAB-antibody-reactive small round bodies with a similar PAB antibody-reactive HW body (arrow) (**c**) can be observed at the paracortical area adjacent to granulomas. Small round bodies (arrows) (**d**) extruding from the large ovoid HW body can be observed in macrophages. All photos are original and were previously published [35]. Scale bar: * 50 μm, ** 1 μm.

**Figure 3 microorganisms-11-00289-f003:**
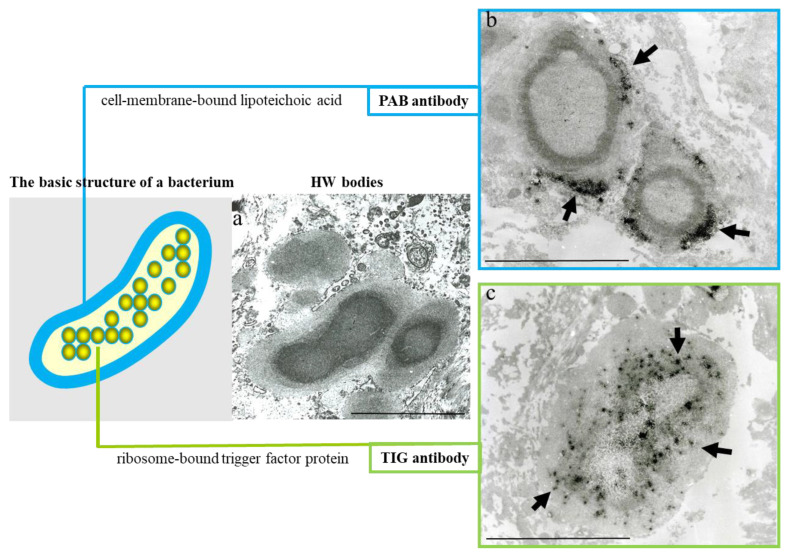
Evidence suggesting that HW bodies are cell-wall-deficient *C. acnes*. HW bodies lack a cell wall structure (**a**). Immunoelectron microscopic localization (black signals indicated by arrows) in HW bodies of two different *C. acnes*-specific bacterial components—cell-membrane-bound lipoteichoic acid detected by the PAB antibody (**b**) and ribosome-bound trigger factor protein detected by the TIG antibody (**c**)—was consistent with their localization in the basic structure of a bacterium despite their lacking a cell wall structure. All photos are original and were previously published [18]. Scale bar: 5 μm.

**Figure 4 microorganisms-11-00289-f004:**
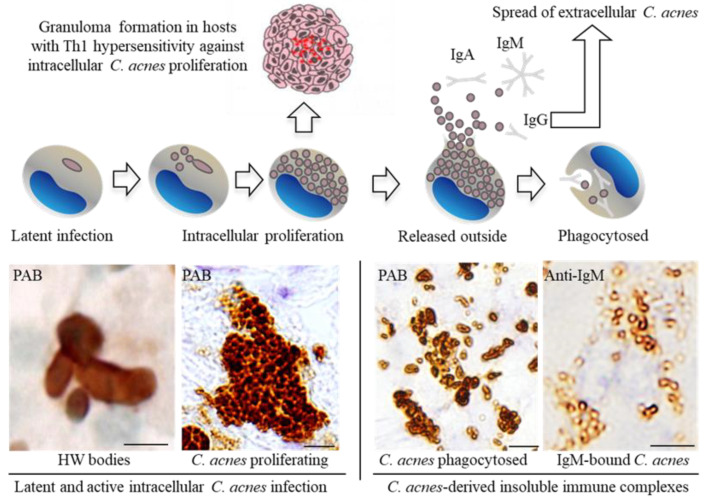
Granuloma formation against the intracellular proliferation of latent *C. acnes* in susceptible hosts and insoluble immune-complex formation with extracellular *C. acnes* outside granulomas after rupture of the infected cells. Representative immunohistochemical features detected by PAB and anti-IgM antibodies are shown below the illustrated macrophages with each stage of intracellular *C. acnes* manifestation. All photos are original and were previously published [35]. Scale bar: 5 μm. HW, Hamazaki-Wesenberg.

**Figure 5 microorganisms-11-00289-f005:**
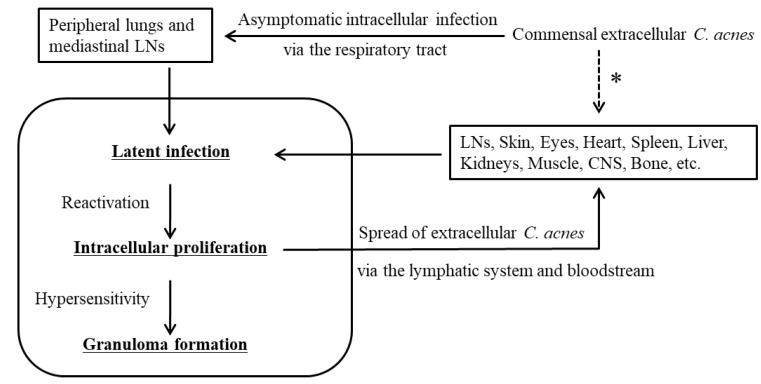
Assumed pathogenesis of sarcoidosis caused by *C. acnes*. Granuloma formation occurs only in some predisposed individuals with Th1 hypersensitivity against the intracellular proliferation of *C. acnes* reactivated at the sites of latent infection. Extracellular *C. acnes* after intracellular proliferation potentially causes new latent infection in the same or other organs. LNs, lymph nodes; CNS, central nervous system. * Local entry and subsequent latent infection of *C. acnes* other than the systemic spread may occur in some organs or patients. The figure is original and was previously published [35].

## Data Availability

The study did not report any data.

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
