# Peer review of "Potential Association of Cutibacterium acnes with Sarcoidosis as an Endogenous Hypersensitivity Infection"

_microorganisms, 2023, doi:10.3390/microorganisms11020289_

Round 1

Reviewer 1 Report

First, I would like to emphasize the comprehensiveness of the review on the state of the art regarding the involvement of C acnes in the pathophysiology of sarcoidosis. It traces the history of the association of this bacterium with the systemic granulomatosis that is sarcoidosis.

One of the major comments would be that the author is in my mind too much focused in C acnes, eluding the possibility that other pathogens or infectious triggers could be associated with sarcoidosis. I also think that some statements about the phylotypes and genetic features of C acnes would be a good point to discuss. 

I also noticed that the author used figures that have already been published without citing the original work (even though the original figures come from Prof. Yoshinobu Eishi).

 I also noticed that nearly 30% of the references are self-citations. Of course, a vast majority of papers on P acnes X sarcoidosis are from YE, but it would be interesting to discuss some other points and also to temper the enthusiasm on C acnes because, in view of the available literature, I think that other infectious triggers (such as fungal triggers for example) should not be forgotten. I was also a bit surprised that the author does not discuss that much the possibility of inter individual differences in the ability to eliminate C acnes (e.g. with autophagy, CWD form of C acnes, endosome/lysosome escape).

Otherwise, this paper summarizes the literature available on the link between C acnes and sarcoidosis quite well.

GENERAL COMMENTS :

First, it seems that it would be appropriate to provide a little more contradictory evidence regarding the involvement of C acnes in human disease. As Mollerup et al. recall in the Journal of Clinical Microbiology (2016, Propionibacterium acnes: Disease-Causing Agent or Common Contaminant? Detection in Diverse Patient Samples by Next Generation Sequencing), C acnes is an extremely common contaminant of human clinical samples, and it is often difficult in routine practice to distinguish between contamination and true infection.

Furthermore, the only available metagenomic study for sarcoidosis patients (with a total sample size of more than 200 patients) found no significant enrichment of C acnes in samples from sarcoidosis patients compared to controls.It is noted that in some samples in this study, however, the Actinomycetota phylum is overrepresented (with possible participation of the Propionibacteriaceae family), but no difference is demonstrated overall (Clarke et al. 2018, AJRCCM) after correction for potential environmental contaminants. Although C. acnes is likely to play a role in the genesis of granulomas in patients with sarcoidosis due to its ability to activate TLR2 and its subjacent signaling pathways, the genesis of the granuloma in relation to this bacterium may be extremely complex, especially given the lack of a blinded study at this time.

Moreover, a certain parallel could be made with other inflammatory granulomatoses such as Crohn's disease, in which the involvement of the microbiome rather than of a particular bacterium has been raised. The existence of a genetic background that alters the clearance of certain pathogens (in particular the alteration of NOD2, a cytosolic sensor for bacterial muramyl peptide, or certain variants of ATG16L1 that could prevent the clearance by autophagy of certain pathogens).

It is known, for example, that C acnes has several phylotypes, some of which are preferentially associated in retrospective studies with certain human pathologies (surgical site infection, sarcoidosis, etc.). On the other hand, some phylotypes are not very infectious or invasive for humans (notably III), and it would have been good to discuss this point concerning sarcoidosis (e.g. Aubin et al. 2017, Anaerobe). Inasmuch as the C acnes genome varies by phylotype and some phylotypes have been more readily associated with sarcoidosis, this additional precision would be interesting, as would discussing bacterial persistence in the host either with intrinsic virulence factors that would depend on the phylotype or bacteriophage carrying, resistance to bacitracin, … (Minegishi et al., Sci Rep, 2015, Scholz et al., Sci Rep, 2016).

COMMENTS IN THE ORDER OF THE MANUSCRIPT

Moreover, here are some comments in the order of the manuscript :

Abstract :

-        Line 8 : “The frequent immunohistochemical detection of Cutibacterium acnes […] ”

o   Since there is no systematic detection of C acnes in sarcoidosis patient samples (most studies are based on Japanese patient samples), I would rather not use the term "frequent".

-        Lines 10-12 : “Histopathologic analysis of sarcoid lymph nodes reveals latent infection and intracellular proliferation of cell wall-deficient C. acnes followed by insoluble immune complex formation.” 

o   Same thing here, I think it would be better to temper this statement since this may not be the case for all sarcoidosis patients.

-        Line 17 : “Many sarcoidosis patients […] ”

o   Same here

Introduction :

-        Lines 32-33 : “[…] environmental factors that trigger the onset of the disease, and pathogens as causative agents […]”.

o   Can the author be more explicit to help the reader distinguish between the second and third points? It seems like pathogens are the environmental trigger in my mind.

o   Moreover, talking about a specific agent triggering sarcoidosis seems premature. Indeed, recent findings from GEWIS studies investigating host-environment interactions have revealed associations of specific triggers in patients with a specific genetic background (e.g., sarcoidosis association with pesticides or smoking in certain subpopulations). I suggest to temper this statement by indicating that some pathogens rather than a particular one could be responsible for sarcoidosis.

Pathology of Granuloma Formation :

o   Lines 57-60 : “If the causative agent is not antigenic […] epithelioid cell granuloma.” I would also modify this statement. In the reference provided by the author, Pagán et al. describe a foreign body granuloma as a granuloma that would be triggered by non-infectious agents (but this can be discussed since Be2+ induced granulomas are more likely to be epithelioid granulomas). In my mind, the proper definition of a foreign body granuloma is a granuloma occurring in the case of an inorganic particle/trigger that does not induce epithelioid cell formation. Also, the proper antigenicity of an environmental trigger is less associated with the granuloma shape/fate than the type of the environmental trigger itself would be.

-        Lines 62-63 : “A pathologic principle […] within the granuloma.”

o   This is still debated for auto-inflammatory disorders like Crohn's disease or Blau syndrome. I suggest to add : “Classically, a pathologic principle …”

The Kveim Reaction :

-        Line 72 : “causative agent”

o   This would be the case if the causative agent is present in the Kveim reagent (Kv). Some proteomic studies of the Kv found that some proteins in the Kv were able to trigger an inflammatory immune response (vimentin (Eberhardt et al. PLoS One 2017),  mKatG from mycobacteria (Song et al, J exp Med, 2005)...). Here, I think that this statement related to the Kv may bring three hypotheses :

§  The Kv tirggers granulomas in predisposed individuals (unregarding of a causative agent but more a panel of antigens than might be able to trigger granulomas)

§  The Kv contain a specific agent or microorganism which is able to trigger granulomas.

§  Both hypotheses

o   Also, some authors provided evidence that the Kv may contain other potential triggers for granulomas that are also found in C acnes cultures (and in some mycobacterias cultures i.e., beta-glucan (Tercelj et al., 2017, Sarcoidosis Vasc Diffues Lung Dis). This would also indicate that potential other triggers can be involved in sarcoidosis onset (here, fungal triggers).

-        I suggest here, again, to temper the statement by avoiding the mention of a unique agent that would be responsible for sarcoidosis onset.

For figures 1, 2 and 3, please cite the original paper from where the figures are extracted.

Histopathologic Analysis of C acnes in Sarcoid Lymph Nodes

-        HW bodies were also described in other diseases (lymphoid malignancies, solid neoplasms, appendicitis, cirrhosis, …) and do not seem to be specific of sarcoidosis. So here, it is difficult to admit that all HW bodies would be cell wall deficient C acnes (CWDCA). Also, HW bodies were also found to be associated in a retrospective series from Alavi et al (Histol Histopathol 1996) to CWD Mycobacterium tuberculosis. So it might be possible that those HW bodies represent CWDB (either C acnes or another bacterium depending on the patient) that would enter on a CWD state to escape the immune system. As an example, L monocytogenes in its CWD form is able to escape the non-activated macrophages’ phagocytosis (Schnell et al. 2014, Front Cell Infect Microbiol). I suggest that the author add a statement indicating that CWDCA is present in sarcoidosis patients’ cells but that this process has not yet been observed for other bacteria or microorganisms.

Pathogenesis of Sarcoidosis Caused by C acnes :

-        Lines 256-7 : “Susceptibility to latent C acnes […] of the host.”

o   NOD receptors are cytosolic sensors to various triggers. To be activated in response to C acnes, NOD1 should be directly in contact with a pattern peptide from C acnes (iE-DAP) or MDP for NOD2 this implicitly suggesting that C acnes material or full C acnes can access the cytosol/escape endosomal degradation. Moreover, the possible defective clearance of C acnes in sarcoidosis patients should be underlined. Calender et al showed in a WES study that there was an accumulation in potentially pathogenic variants in autophagy pathway in sarcoidosis patients (in a similar way than in Crohn's disease, where gram negative bacteria clearance is also impaired (through ATG16L1 polymorphisms for example)), so here, the possible link between the inability of the host to get rid of the pathogen should be underlined. There is a possibility that if autophagy is one of the major ways of eliminating C acnes, sarcoidosis patients with potentially defective autophagy cannot get rid of the pathogen. So here, I suggest to discuss/mention the possibility that sarcoidosis patients are unable to get rid of some pathogens (e.g., C acnes) and how (like what is discussed in the paper from Nakamura et al., for example with a statement on C acnes hemolysin CAMP).

Treatment strategies for Refractory sarcoidosis :

-        Lines 310-2 : “Combination therapy […] cardiac sarcoidosis.”

o   Here the author should also mention the pending clinical trial of anti C acnes antimicrobial therapy in sarcoidosis (i.e., PHENOSAR study (NCT05291468))

Allergic Endogenous Infection Caused By C acnes :

-        Line 322 : “Pneumocystis carinii”:

o   The denomination carinii is outdated, please use P jiroveci instead.

-        Lines 324-30 : “Sarcoidosis caused […] endogenous infections.”

o   The use of the term "allergic" seems here (and in more generally along the manuscript) improper to what could happen in sarcoidosis. Indeed, sarcoidosis patients may have :

§  former contact with the causative antigen this inducing immunization and/or creating immune complexes

§  exagerated immune reaction to the pre cited antigen

o   But this could maybe refer more to the type III reaction for the Gell and Coombs classification (hypersensitivity with immune complexes) rather than the type I "real" allergic reaction (since no IgE or proper histamine mediated immune response is involved so far in sarcoidosis).

o   Here, I suggest to replace “allergy” with “hypersensitivity”. This is also suggested for the title of the manuscript as well.

Reviewer 2 Report

The manuscript concerns important and interesting topic and gains new knowledge, describes the hypothesis of sarcoidosis as an allergic endogenous infection. This manuscript deal with the role of host immunology and components of C. acnes in developing sarcoidosis. The text is well written however some chapters seems to be very short.

My major remarks to this manuscript is that near all figures presented in were already published:

Fig.1.   Fig. 2C,D,E,F J Clin Med. 2021

Fig.2 – Fig.3 J Clin Med. 2021

Fig.3. – similar to Fig.12 BioMed Research International 2013

Fig.4 – Fig.5 J Clin Med. 2021

Fig.5 –  Fig.6 J Clin Med. 2021, and Fig. 23 BioMed Research International 2013

Fig. 6 – Fig. 22 BioMed Research International 2013

To avoid any criticism it should be clearly stated f.ex. “figure 1 are reproduced from…. with permission".

Minor remarks

p.1.28   granulomas are formed

p.1. 31 “three” instead of 3

p.2. 54 issues “still” remain

p.2.  76 “two” instead of 2

p.6. 215 “two” instead of 2

p.7. 245-247      the limitations of the novel method should be also mentioned

p.9. 322-324 remove the sentence about P. carinii and remove fig.6

p.9 326-329 remove this sentence corresponding to figure 6 accordingly.

p.9 paragraph 13

I propose to transfer this paragraph on the end of the manuscript – treatment strategies as the last subject.

p.12. 446 add journal

p.12. 450 add journal

p.12. 475 add bibliographic data

p.14. 574 add journal
